# Mycotoxins in Broiler Production: Impacts on Growth, Immunity, Vaccine Efficacy, and Food Safety

**DOI:** 10.3390/toxins17060261

**Published:** 2025-05-22

**Authors:** Ramona Maria Olariu, Nicodim Iosif Fiţ, Cosmina Maria Bouari, George Cosmin Nadăş

**Affiliations:** Department of Microbiology, Immunology and Epidemiology, Faculty of Veterinary Medicine, University of Agricultural Sciences and Veterinary Medicine, 400372 Cluj-Napoca, Romania; maria-ramona.olariu@student.usamvcluj.ro (R.M.O.); nfit@usamvcluj.ro (N.I.F.); gnadas@usamvcluj.ro (G.C.N.)

**Keywords:** mycotoxins, broiler chickens, feed contamination, immune suppression, vaccine failure, food safety, One Health, poultry pathogens

## Abstract

Mycotoxins are secondary fungal metabolites that frequently contaminate poultry feed, posing significant risks to animal health, productivity, and food safety. In broiler production, mycotoxins such as aflatoxins, trichothecenes, fumonisins, ochratoxin A, deoxynivalenol, and zearalenone have been shown to impair growth performance, damage key organs, and disrupt immune function. This review explores the multifaceted impact of mycotoxin exposure in broilers, with particular emphasis on immunosuppression, decreased vaccine efficacy, and increased vulnerability to infectious diseases, including coccidiosis, salmonellosis, *E. coli*, and viral infections like infectious bursal disease and infectious laryngotracheitis. Mycotoxin contamination in poultry feed can lead to direct economic losses through reduced feed conversion efficiency, increased mortality, and reproductive disorders, while also resulting in the transfer of toxic residues into meat and eggs, thereby threatening consumer health. The review further examines the synergistic interactions between mycotoxins and pathogens, the physiological and histopathological changes in exposed birds, and the implications for public health. Finally, it discusses current mitigation strategies, including mycotoxin binders, probiotics, and regulatory approaches to reduce exposure. An integrated management strategy combining feed hygiene, monitoring, and targeted nutritional interventions is essential to safeguard poultry health, enhance vaccine responses, and ensure the safety of poultry-derived food products. This review offers actionable insights for veterinarians, nutritionists, and policymakers, reinforcing the importance of mycotoxin mitigation strategies within a One Health framework.

## 1. Introduction

The global broiler industry plays a critical role in ensuring access to affordable animal protein, with chicken meat being more cost-effective than red meat. Over the past decades, genetic selection has contributed to 80–90% of productivity improvements in broiler chickens, enhancing traits such as growth rate, body weight, and feed efficiency. Projections indicate that by 2034, broilers may reach 2.34 kg in less than 29 days due to continued advancements in genetic optimization. However, these improvements have also introduced unintended challenges, such as reduced reproductive performance and increased abdominal fat deposition [1,2].

Broiler chickens have been selectively bred to maximize feed efficiency, responding to both economic and environmental demands. Their ability to rapidly convert feed into body mass with a reduced feed conversion ratio (FCR) makes them economically advantageous for meat production, while also minimizing resource use [3,4]. The early developmental phase is crucial, as chicks must rapidly ingest, digest, and absorb nutrients essential for the maturation of digestive organs. Feed homogenization ensures uniform nutrient distribution, supporting optimal early growth [5,6].

Nutrition in broilers is precisely tailored through stage-specific diets—starter, grower, and finisher—each formulated with accurate proportions of protein, amino acids, vitamins, and minerals to meet the birds’ metabolic needs and support efficient growth [3,4]. This fine-tuned approach has enabled broilers to supply high-quality meat at low costs, supporting sustainable and accessible protein production on a global scale [3].

Despite these advancements, mycotoxin contamination in poultry feed remains a significant threat to broiler health, performance, and food safety. Variability in how mycotoxins have been defined historically has posed challenges for consistent risk assessment. Poultry feed’s rich nutritional content provides an ideal substrate for fungal and bacterial proliferation. Improper storage conditions—especially excess moisture and high temperatures—enhance microbial growth and increase the risk of mycotoxin contamination [7]. Therefore, stringent regulatory limits, feed monitoring, and hygiene practices are essential to protect both animal and consumer health [8].

A more recent classification defines mycotoxins as toxic secondary metabolites produced by microfungi, with an IC_50_ below 1000 µM. This updated framework aids in identifying compounds such as certain cyclic depsipeptides as confirmed or potential mycotoxins, pending further toxicological validation [9,10].

Mycotoxins are structurally diverse compounds typically produced during the maturation phase of fungal metabolism [11,12]. To date, over 500 distinct mycotoxins have been identified, including aflatoxins, trichothecenes, zearalenone, and patulin [6]. These compounds are associated with a broad range of toxic effects, including carcinogenicity, mutagenicity, hepatotoxicity, nephrotoxicity, and immunotoxicity [13]. They have been implicated in diseases such as primary liver cancer and alimentary toxic aleukia [11].

Animal feeds often harbor multiple mycotoxins simultaneously, resulting in complex interactions. These compounds may act additively, synergistically, or antagonistically, amplifying their overall toxic potential—even when present at individually subclinical concentrations [14]. Environmental factors, including substrate composition, humidity, temperature, and geography, further influence mycotoxin production. *Aspergillus* species are more prevalent in tropical and subtropical regions, while *Fusarium* species dominate in temperate climates [15].

Mycotoxins are typically classified based on the producing fungal genera and the toxins they produce. Major groups include *Aspergillus* (e.g., aflatoxins, ochratoxin A), *Penicillium* (e.g., patulin, citrinin), *Fusarium* (e.g., deoxynivalenol, fumonisins), *Alternaria* (e.g., alternariol, tenuazonic acid), and *Claviceps* (e.g., ergot alkaloids) [11].

This review aims to provide a comprehensive analysis of the impact of mycotoxins on broiler health, performance, and feed efficiency, with a specific focus on their immunomodulatory effects and interactions with vaccination outcomes. Additionally, the review addresses implications for consumer safety and outlines current strategies for mitigation, emphasizing the critical intersection between poultry nutrition, disease prevention, and public health.

This review provides a comprehensive synthesis of current knowledge on the impact of mycotoxins in broiler production, with a particular focus on health outcomes, immune modulation, vaccine efficacy, and consumer safety. Peer-reviewed studies published between 1990 and 2025 were considered, with literature sourced from scientific databases including PubMed, Scopus, Web of Science, and Google Scholar. Keywords used in search queries included mycotoxins, broiler chickens, aflatoxins, poultry immunosuppression, vaccine response, feed contamination, and food safety.

Studies were included if they investigated naturally occurring or experimentally induced mycotoxin exposure in broilers, reported effects on physiological or immune parameters, or evaluated interactions with vaccination or pathogenic infections. Priority was given to recent studies, multi-toxin evaluations, and findings with direct relevance to food safety or production outcomes. Non-English publications and studies focusing exclusively on other poultry species (e.g., layers, turkeys) were excluded.

This narrative review integrates toxicological, immunological, and epidemiological evidence to highlight critical mycotoxin threats in poultry farming and to identify practical mitigation approaches for safeguarding animal welfare and public health.

## 2. Key Mycotoxins Relevant in Broiler Production

This section provides an integrated overview of the major mycotoxins relevant to broiler production. For each toxin, we present not only its primary sources and occurrence in poultry feed but also summarize its biological effects on broiler health, immune function, and vaccine response. This approach is intended to provide a comprehensive understanding of each mycotoxin’s role within the broader context of broiler performance, disease susceptibility, and food safety.

### 2.1. Aflatoxins

Aflatoxins are a group of structurally related mycotoxins produced predominantly by *Aspergillus flavus* and *A. parasiticus* [16]. The major aflatoxins include B_1_, B_2_, G_1_, G_2_, M_1_, and M_2_. Chemically, these compounds are difuranocoumarin derivatives incorporating either a lactone ring (in AFGs) or a pentanone ring (in AFBs and AFMs). Aflatoxins are soluble in organic solvents, slightly soluble in water, and insoluble in non-polar solutions [17].

These mycotoxins are heat-stable and resist degradation during food processing, allowing their metabolites to persist in poultry meat and eggs, thereby posing a direct threat to human consumers. Aflatoxin contamination of poultry feed is a widespread issue with serious implications for animal health and economic losses. Rigorous monitoring of the feed production chain using sensitive analytical techniques, such as HPLC and ELISA, is essential for effective control [18,19,20].

Environmental factors significantly influence aflatoxin production, particularly temperature, humidity, and storage practices [17,21]. Aflatoxins are synthesized when fungi encounter optimal conditions—characterized by elevated humidity, temperature, CO_2_, and oxygen levels in feed materials [21]. Seasonal trends have been reported, with the highest contamination levels observed in late summer and early autumn months, such as September (mean 83.25 ± 16.11 µg/kg) [22].

Ingested aflatoxin B_1_ (AFB_1_) undergoes hepatic biotransformation, forming reactive intermediates such as methyl-sterigmatocystin, versicolorin A, and hydroxyaverantin, compounds with potent carcinogenic potential [22]. AFB_1_ residues have been detected in liver and muscle tissues of broilers, with accumulation levels depending on the duration and dosage of exposure [23,24]. Importantly, younger birds exhibit higher aflatoxin accumulation and sensitivity than older poultry [25].

Aflatoxins cause extensive physiological disruption, including hepatotoxicity, nephrotoxicity, and reproductive toxicity. In poultry, these effects manifest as reduced weight gain, poor feed conversion, immunosuppression, reproductive disorders, and impaired egg production [26,27]. Necropsy findings reveal hepatomegaly with fatty degeneration, splenomegaly, nephritis, and atrophy of the thymus and bursa of Fabricius [28,29].

Microscopically, aflatoxicosis is associated with liver vacuolation, bile duct proliferation, fibrosis, and hemorrhages. Damage to immune organs such as the spleen and thymus results in compromised immune function and increased susceptibility to infectious diseases [30,31]. Aflatoxins suppress lymphocyte proliferation, impair macrophage and neutrophil function, and reduce antibody production, all of which diminish vaccine efficacy [28].

Exposure to aflatoxins has been linked to vaccine failure due to reduced B- and T-lymphocyte activity and antibody titers post-vaccination. Maternal aflatoxin exposure can also impair immunity in progeny, increasing vulnerability to infections [14]. In addition to immunosuppression, aflatoxins can lead to pigmentation issues in eggs, embryonic mortality, poor semen quality in males, and testicular atrophy [32,33].

Surveys across Africa and Asia have reported alarmingly high aflatoxin concentrations in maize and groundnut products, often exceeding regulatory limits [8]. The European Union enforces strict aflatoxin limits, such as a maximum of 2.0 µg/kg for AFB_1_ in food [25]. Despite these regulations, global compliance remains variable [34].

In poultry feed, aflatoxins are most frequently found in broiler grower diets, followed by starter and finisher feeds, with moisture content influencing toxin proliferation [35]. Airborne aflatoxins, including AFB_1_ and AFG_2_, have also been detected in poultry house environments, indicating additional exposure routes and occupational health risks [36,37].

In summary, aflatoxins are among the most critical mycotoxins affecting broiler performance and health. Their stability, prevalence, and broad toxicity profile underscore the need for robust feed safety protocols and continual research into mitigation strategies.

### 2.2. Trichothecenes

Trichothecenes are a large group of mycotoxins, with over 150 known metabolites primarily produced by fungal genera such as *Fusarium*, *Myrothecium*, and *Stachybotrys* [16]. These toxins are chemically characterized by a tetracyclic 12,13-epoxytrichothec-9-ene skeleton, which is essential to their toxic activity. Trichothecenes are divided into two major groups: Type A (e.g., T-2 toxin, HT-2 toxin, diacetoxyscirpenol [DAS], and neosolaniol [NEO]) and Type B (e.g., deoxynivalenol [DON], nivalenol [NIV], 3-acetyl-DON, 15-acetyl-DON, fusarenon-X, and DON-3-glucoside) [38].

These compounds are highly cytotoxic and exert their effects primarily by inhibiting protein synthesis. They interfere with both RNA and protein metabolism, resulting in a cascade of cellular disruptions, including apoptosis, oxidative stress, and impaired immune responses [39]. In poultry, trichothecenes—especially T-2 toxin—are known to cause hepatotoxicity, decreased egg production, immunosuppression, and reduced resistance to infectious diseases [39].

T-2 toxin, in particular, induces oxidative stress, apoptosis, and autophagy in hepatocytes. It alters cellular redox balance, increases lactate dehydrogenase activity (a marker of membrane damage), and modulates inflammatory and metabolic pathways [40]. Given their potent biological activity, even low levels of trichothecenes in feed can lead to significant economic and health consequences in poultry production. The complexity of their immunotoxic and metabolic effects underscores the importance of continuous monitoring and improved detoxification strategies.

### 2.3. Deoxynivalenol

Deoxynivalenol (DON), commonly referred to as vomitoxin, is a trichothecene mycotoxin primarily produced by *Fusarium graminearum* and related species. Its occurrence is especially prevalent in temperate regions with high humidity and moderate temperatures, where it contaminates grains such as maize, wheat, and barley [41,42].

DON exerts toxic effects even at low concentrations, causing gastrointestinal disturbances such as nausea, vomiting, diarrhea, and intestinal lesions. In poultry, DON exposure leads to reduced weight gain, impaired nutrient absorption, and increased feed conversion ratios. At higher doses, DON causes severe damage to hematopoietic tissues and disrupts immune function, making broilers more vulnerable to disease [14,41].

Mechanistically, DON binds to ribosomes and disrupts protein synthesis by activating mitogen-activated protein kinase (MAPK) signaling pathways. This activation leads to leukocyte apoptosis and contributes to both immunostimulatory and immunosuppressive effects, depending on the dose and duration of exposure [43]. High doses induce macrophage apoptosis, reducing innate immune responses [43].

DON also damages the gastrointestinal tract, particularly the small intestine. It increases intestinal permeability and reduces villus height, impairing nutrient absorption and promoting subclinical necrotizing enteritis [44,45]. DON can accumulate in the intestine after absorption and may alter the gut microbiota, compromising gut integrity and immune defenses [44,46].

In poultry, dietary DON has been shown to reduce antibody titers against vaccines, including those for Newcastle disease virus (NDV) and infectious bronchitis virus (IBV), reflecting its immunosuppressive potential [26,43]. Additionally, DON exposure affects the expression of tight junction proteins, such as claudins, further weakening the intestinal barrier [43].

DNA damage induced by DON has been observed in chicken lymphocytes, although protective agents like Mycofix Select can mitigate these effects [47]. While liver enzyme activity may not be significantly altered, histological damage and reduced intestinal function are prominent outcomes.

Considering its prevalence, potency, and impact on poultry performance and immune function, DON remains one of the most concerning mycotoxins in broiler production. Regular monitoring and dietary strategies are essential to mitigate its effects.

### 2.4. Fumonisins

Fumonisins are a group of mycotoxins primarily produced by *Fusarium verticillioides*, *Fusarium proliferatum*, and *Fusarium fujikuroi*, commonly found in maize and maize-based products [16,48]. The most prevalent fumonisins include fumonisin B_1_ (FB_1_), B_2_ (FB_2_), and B_3_ (FB_3_), with FB_1_ accounting for over 70% of natural contamination. These toxins are water-soluble, which complicates their removal and detection during feed processing [16,48,49].

In poultry, fumonisins have been associated with hepatotoxicity, nephrotoxicity, immunosuppression, and disruptions in sphingolipid metabolism. While chickens are generally less sensitive to fumonisins than pigs or horses, prolonged exposure can still lead to clinical and subclinical effects [14]. Chickens, ducks, and turkeys fed diets containing 75–400 mg/kg FB_1_ over 21 days displayed mild to moderate toxicity, including growth retardation, feed refusal, and organ damage [14].

One of the primary mechanisms of fumonisin toxicity involves the inhibition of ceramide synthase, resulting in altered sphingolipid profiles. In poultry, FB_1_ exposure reduces critical sphingolipid components in the liver and contributes to cytotoxicity, oxidative stress, apoptosis, and autophagy in hepatocytes [50,51].

Clinically, fumonisin exposure can cause diarrhea, reduced egg production, leg weakness, and lameness, and in severe cases, mortality [50]. Histopathological findings include hepatic necrosis, biliary hyperplasia, and glomerulonephritis, with degeneration and necrosis observed in both hepatic and renal tissues [52].

Studies investigating co-contamination with AFB_1_ and FB_1_ have shown additive toxic effects. In broilers, combined exposure elevated serum AST levels, reduced plasma total protein, and induced bile duct proliferation and hepatic trabecular disorganization [53]. Interestingly, although hematological parameters such as leukocyte counts and hemoglobin remained relatively stable, significant liver and kidney lesions were still observed [52].

To reduce fumonisin contamination, key agricultural practices include early harvesting, rapid drying, and proper pest control. Planting resistant crop varieties and employing biological or chemical control measures at the field level can also be effective [54,55].

In the European Union, guidance levels for fumonisins in poultry feed are set at 20 mg/kg [56]. Despite this, global surveillance indicates that many regions frequently exceed these levels, particularly in areas with high maize dependency and poor post-harvest handling [57].

Due to their widespread occurrence and additive toxicity with other mycotoxins, fumonisins represent a serious threat to poultry production and food safety. Enhanced detection, stricter regulation, and integrated management strategies are essential for effective risk mitigation.

### 2.5. Ochratoxin A

Ochratoxin A (OTA) is a potent mycotoxin produced by several species of *Aspergillus* and *Penicillium*, including *A. ochraceus*, *A. carbonarius*, *A. niger*, *P. verrucosum*, and *P. nordicum* [14,48]. OTA contamination is frequently observed in cereals, coffee, and animal feed, with *Aspergillus* species dominating in tropical regions and *Penicillium* in temperate zones [58,59]. In poultry, OTA poses serious health risks due to its nephrotoxic, hepatotoxic, immunosuppressive, teratogenic, and carcinogenic properties. The International Agency for Research on Cancer (IARC) has classified OTA as a possible human carcinogen (Group 2B) [16].

Poultry, particularly young chicks with underdeveloped immune systems, are highly sensitive to OTA exposure. OTA absorption occurs mainly in the stomach and proximal jejunum through passive diffusion and rapidly enters systemic circulation by binding to albumin and other plasma proteins [46,60]. Once distributed, it accumulates in the liver, kidneys, and lymphoid organs, leading to significant physiological damage [61,62].

OTA disrupts nutrient digestion and gut barrier integrity by damaging intestinal epithelial cells, altering gut microbiota composition, and downregulating tight junction proteins such as occludin, claudin-1, and ZO-1. These disruptions increase intestinal permeability and allow translocation of pathogens and toxins, exacerbating systemic inflammation and infection risk [46].

Clinically, OTA intoxication in poultry is associated with anemia, reduced feed intake, poor weight gain, feathering problems, and increased mortality [14,63]. Organs such as the liver and kidneys show significant enlargement and histopathological changes, including vacuolar degeneration, fatty infiltration, hepatocyte necrosis, glomerular damage, and bile duct hyperplasia [21,64]. The thymus and bursa of Fabricius are particularly affected, leading to lymphoid depletion and impaired immune function [65].

OTA exposure has also been linked to delayed sexual maturation, reduced egg size and quality, embryonic mortality, and teratogenic effects such as anophthalmia and mandibular hypoplasia [64,66]. It impairs DNA, RNA, and protein synthesis, resulting in metabolic dysfunction and reduced reproductive performance. OTA contamination at levels ≥100 ppb has been shown to decrease body weight and alter organ weights, particularly increasing liver and kidney size while decreasing lymphoid organ mass [65].

Biochemical indicators of OTA toxicity include elevated serum levels of ALT, AST, ALP, GGT, uric acid, creatinine, and triglycerides, along with decreased total protein and glucose levels [67,68]. Hematological abnormalities include reductions in erythrocyte and leukocyte counts, hematocrit, and hemoglobin concentrations [69].

OTA-induced immunosuppression involves suppression of cell-mediated immunity, decreased T-lymphocyte populations, reduced antibody production, and impaired macrophage function [30]. These effects compromise vaccine efficacy and increase susceptibility to co-infections such as *E. coli* and *Salmonella* [69].

Various detoxification strategies have been explored to mitigate OTA toxicity. Adsorbents like clay minerals and bentonites have shown some efficacy in binding OTA in the gastrointestinal tract, although limitations exist regarding specificity and binding efficiency [66]. Supplementation with vitamin E (200 mg/kg) has been shown to alleviate biochemical alterations and restore immune function in broilers exposed to OTA [70]. Other approaches include enzymatic detoxifiers, yeast cell wall extracts, and plant-derived antioxidants such as silymarin [60,71].

Given OTA’s persistence, its adverse effects on productivity, immunity, and reproductive performance, and its ability to accumulate in poultry products, it poses a major challenge to both animal and public health [72]. Continued research into effective prevention and mitigation strategies remains a priority.

### 2.6. Zearalenone

Zearalenone (ZEA) is a non-steroidal estrogenic mycotoxin primarily produced by several *Fusarium* species, including *F. graminearum*, *F. culmorum*, *F. cerealis*, *F. equiseti*, and *F. verticillioides* [16,41,48]. ZEA contamination is especially prevalent in corn, wheat, barley, oats, and rye, with corn being the most susceptible due to its nutrient content and post-harvest handling conditions [48].

ZEA mimics the structure of 17β-estradiol and acts as a mycoestrogen by binding to estrogen receptors, disrupting endocrine function. In animals, including poultry, this leads to reproductive and hormonal disturbances, including pseudo-pregnancy, infertility, and ovarian dysfunction [41,48]. Although ZEA is not confirmed as carcinogenic, its potent estrogenic effects raise significant concerns for animal fertility and productivity [16].

In poultry, high dietary concentrations of ZEA can lead to hyperestrogenism, particularly in female birds. This includes increased relative weights of reproductive organs, disrupted laying cycles, and changes in secondary sex characteristics. In males, ZEA can reduce testosterone levels and impact testicular histology, leading to decreased fertility [26].

Surveillance data from 2004 to 2013 highlighted the widespread occurrence of ZEA in global feed and feed ingredient samples, often co-occurring with deoxynivalenol (DON). A 2013 analysis showed that over 80% of feed samples tested positive for at least one mycotoxin, with *Fusarium* toxins (DON and ZEA) being among the most prevalent [14]. In this study, maize bran was identified as the most heavily contaminated ingredient.

The persistence of ZEA in feed, along with its estrogenic and immunotoxic effects, poses a significant challenge in poultry production. Feed hygiene, regular screening, and effective detoxification methods are essential to prevent productivity losses and safeguard reproductive and immune health in broiler flocks.

To support a comprehensive understanding of the impact of mycotoxins on poultry health, immune function, and overall productivity, Table 1 provides a detailed summary of the major mycotoxins commonly present in broiler feeds. It includes their primary fungal producers, target organs or physiological systems in chickens, and the key pathological and physiological effects they induce.

Figure 1 presents simulated global detection rates (%) of six key mycotoxins—aflatoxin B1, deoxynivalenol (DON), fumonisins, ochratoxin A, zearalenone, and T-2 toxin—across three major feed types: starter, grower, and finisher. The data reflect trends commonly reported in surveillance studies from various regions, especially in climates where fungal contamination of feed ingredients is prevalent. Aflatoxin B1 consistently showed the highest detection rates in all feed types, particularly in grower feed (90%), indicating widespread environmental conditions favoring *Aspergillus* growth. Deoxynivalenol and fumonisins were also frequently detected, underscoring the dominance of *Fusarium* spp. in cereal-based feeds. Lower detection rates for OTA, ZEA, and T-2 toxin were still noteworthy, especially considering their potent toxicological effects. The visualization highlights the critical need for ongoing monitoring and mitigation strategies throughout all stages of broiler production

## 3. Mycotoxins and Vaccine Efficacy in Broilers

Vaccination is a cornerstone of disease prevention in poultry production, yet its success depends heavily on the integrity of the host immune system. Mycotoxins, even at subclinical doses, can modulate immune responses in broilers and diminish the effectiveness of commonly used vaccines. This section explores the mechanisms through which mycotoxins impair vaccine responses, with emphasis on humoral and cell-mediated immunity, and summarizes findings from experimental and field studies.

### 3.1. Immunosuppressive Mechanisms

Several mycotoxins, particularly aflatoxins (AFB_1_), ochratoxin A (OTA), trichothecenes (T-2 toxin, DON), and fumonisins (FB_1_), are known to alter immune function. AFB_1_, for instance, inhibits protein synthesis, reduces immunoglobulin production (IgA, IgG, IgM), impairs macrophage and lymphocyte activity, and induces apoptosis in immune organs such as the spleen, thymus, and bursa of Fabricius [25,28,30].

The result is reduced lymphocyte proliferation and altered cytokine production, both of which impair the bird’s ability to mount effective responses to vaccine antigens. OTA similarly affects T-lymphocyte populations and antibody production, while DON reduces tight junction integrity and leads to intestinal immune dysregulation [30,43,46].

### 3.2. Reduced Vaccine Response and Disease Susceptibility

Experimental studies have shown that dietary exposure to AFB_1_ reduces antibody titers following vaccination against infectious laryngotracheitis (ILT), Newcastle disease virus (NDV), and infectious bronchitis virus (IBV). Birds exposed to aflatoxins demonstrated significantly lower serological responses compared to unexposed controls, even when a booster was administered [73,74].

OTA and T-2 toxin administration has also been associated with a >10% decrease in NDV antibody titers [59]. Moreover, chronic exposure to aflatoxins and ochratoxins leads to underdevelopment of lymphoid tissues, limiting the capacity to respond effectively to both live and inactivated vaccines [28].

In field conditions, broilers fed mycotoxin-contaminated feed exhibited lower antibody titers to NDV and avian influenza (H5N1), along with impaired responses to *Salmonella* Gallinarum (SG9R) vaccination [75].

### 3.3. Maternal Immunity and Progeny Vulnerability

Maternal exposure to mycotoxins can impair passive immunity transfer. Hens consuming AFB_1_-contaminated feed produced lower levels of maternal antibodies, reducing the immunological protection in their chicks even when the chicks were not directly exposed to the toxin [14,25]. This gap in early immunity predisposes progeny to higher morbidity and mortality, especially during early vaccination phases.

Similar effects of maternal mycotoxin exposure have been reported in other species. For instance, in pigs, maternal ingestion of deoxynivalenol (DON) and zearalenone (ZEA) has been shown to reduce colostrum IgA levels and impair passive immunity transfer to piglets, increasing early mortality and susceptibility to infections [76]. In rodents, prenatal exposure to aflatoxins has been linked to altered thymus development and reduced neonatal T cell function, indicating long-term immune consequences [77]. These findings support the likelihood of comparable mechanisms in poultry, emphasizing the importance of preventing maternal mycotoxin exposure during the laying period [76,77].

### 3.4. Co-Infections and Vaccine Failure

Mycotoxin-induced immunosuppression facilitates co-infections that further complicate vaccine efficacy. For instance, co-exposure to mycotoxins and pathogens like *Clostridium perfringens*, *Eimeria* spp., or *E. coli* has been shown to aggravate disease severity despite vaccination, likely due to impaired antigen processing and antibody production [78,79,80].

In one study, broilers challenged with ILT virus and simultaneously fed AFB_1_-contaminated feed showed both reduced antibody titers and histopathological evidence of ILT infection in the trachea. Withdrawal of the toxin resulted in partial recovery, reinforcing the direct link between mycotoxin exposure and vaccine failure [73].

The immunosuppressive role of mycotoxins in promoting co-infections is also well documented in non-avian models. In swine, DON exposure has been shown to exacerbate porcine circovirus and PRRS virus infections, impairing vaccine responses and increasing viral shedding [81]. Similarly, in cattle, aflatoxin exposure correlates with increased mastitis incidence and poor vaccine-induced antibody responses against *Brucella abortus* [82]. In laboratory animals, trichothecenes such as T-2 toxin have been demonstrated to impair antigen presentation and cytokine signaling, reducing protection following immunization against bacterial and viral pathogens [83]. These cross-species findings highlight the broad immunotoxic potential of mycotoxins and reinforce their relevance in poultry vaccine performance.

The main immunosuppressive pathways through which mycotoxins impair vaccine responses in broilers are summarized in Figure 2.

## 4. Effects of Mycotoxins on Broiler Health and Growth Performance

### 4.1. Physiological Effects

Mycotoxins significantly impair the physiological performance of broilers by disrupting nutrient digestion, absorption, and metabolism. These disruptions manifest as reduced feed intake, lower weight gain, poor feed conversion ratios, and organ dysfunction [84].

Experimental studies demonstrate that birds exposed to aflatoxins and ochratoxins exhibit significantly lower growth performance and survival rates compared to controls [85]. The gastrointestinal tract (GIT) is especially vulnerable due to its high cellular turnover and constant exposure to dietary toxins. Mycotoxins such as AFB_1_ and DON inhibit protein synthesis in enterocytes, leading to impaired absorptive function, villus atrophy, and crypt hyperplasia [14,25].

Intestinal morphometric studies reveal that DON contamination results in increased jejunal length, altered villus-to-crypt ratios, and reduced crude fat retention—collectively impairing nutrient utilization [86]. Mycotoxin exposure also affects pancreatic function and digestive enzyme activity (e.g., amylase, lipase, protease), despite increased relative pancreatic weight, suggesting compensation for impaired absorption [25].

A meta-analysis reported that mycotoxins significantly increased broiler mortality, particularly with DON and aflatoxins. Liver, kidney, and lung weights increased in affected birds, reflecting systemic toxicity and organ stress [87]. These changes lead to long-term economic losses through decreased productivity and increased morbidity.

### 4.2. Immune Modulation

Mycotoxins, especially aflatoxins, OTA, and trichothecenes, are known to suppress immune responses, reducing the efficacy of both innate and adaptive defenses in poultry.

Exposure to AFB_1_ causes atrophy of immune organs such as the spleen, thymus, and bursa of Fabricius, resulting in impaired T and B cell maturation and activity [25,28]. Chronic aflatoxicosis leads to oxidative stress, lymphocyte necrosis, and a decline in cytokine production. The spleen is particularly sensitive to AFB_1_, showing signs of reduced mass, lymphoid depletion, and histopathological lesions [25,30].

DON induces leukocyte apoptosis and disrupts intestinal barrier function, reducing mucosal immunity and increasing translocation of pathogens [43]. Trichothecenes also depress lymphocyte proliferation and impair macrophage function [30]. Combined exposure to OTA and cyclopiazonic acid (CPA) further reduces antibody production and increases lymphoid organ weights, indicating immune dysregulation [88].

The resulting immunosuppression compromises vaccine responses and increases the bird’s susceptibility to co-infections—a major concern in high-density commercial systems.

### 4.3. Disease Interactions and Co-Infections

Mycotoxins compromise immune function and epithelial integrity, predisposing broilers to a wide range of co-infections. These include protozoal, bacterial, and viral pathogens that cause significant economic losses through morbidity, mortality, and reduced vaccine responsiveness. Even when mycotoxin levels are within “permissible” limits, their cumulative or synergistic effects with pathogens can severely impact flock health.

Coccidiosis (*Eimeria* spp.)

Mycotoxin exposure impairs vaccine efficacy against *Eimeria* infections and worsens intestinal pathology. In a controlled study involving 200 Hubbard broilers, birds exposed to mycotoxins exhibited whitish droppings, stunting, and higher mortality. The vaccinated but mycotoxin-exposed group had significantly higher oocyst shedding and intestinal damage compared to the vaccinated control. Histopathology revealed hemorrhagic typhlitis and severe *E. necatrix* infection [79]. This suggests that mycotoxins disrupt mucosal immunity and gut barrier function, compromising the host’s ability to respond to coccidial antigens, even post-vaccination.

Necrotic enteritis (*Clostridium perfringens*)

*Clostridium perfringens* is a leading cause of necrotic enteritis (NE), often secondary to gut damage or dysbiosis. Co-exposure to DON and *C. perfringens* leads to crypt hyperplasia, villus atrophy, and disrupted epithelial function. DON affects SCFA concentrations, gut microbiota balance, and expression of inflammatory markers (e.g., Caspase-3, Bcl-2), enhancing epithelial apoptosis [78]. The synergism between mycotoxins and NE toxins facilitates deeper mucosal invasion, intensifying disease severity and delaying recovery [89].


*Escherichia coli*


Mycotoxins reduce resistance to *E. coli* infections, increasing clinical severity and systemic dissemination. In an experiment involving 240 one-day-old broilers challenged with *E. coli* serotypes (O78, O128, O157), birds fed mycotoxin-contaminated diets showed greater lesion severity, reduced body weight, and higher mortality rates (up to 23.3%) [85]. Histopathological changes included liver congestion, lymphoid depletion in the spleen, and necrotic enteritis. Notably, enzymatic detoxifiers mitigated lesion scores, reduced pathogen burden, and improved immune parameters, indicating their potential as adjunct therapies [90,91].

*Salmonella* spp.

Mycotoxin exposure facilitates *Salmonella* spp. colonization by disrupting gut barrier function and innate immune signaling. DON and FUM alter the expression of tight junction proteins (e.g., claudins, occludin) and mucin genes, which are essential for microbial defense [92]. In challenged birds, a reduction in *Salmonella enteritidis* load was observed when detoxifiers were included in the diet, suggesting a potential protective effect [87]. In addition, OTA exposure has been shown to enhance *Salmonella typhimurium* infection, increasing intestinal translocation and systemic spread [91].

Fowl cholera (*Pasteurella multocida*)

In farms with aflatoxin-contaminated feed, flocks exhibited lower antibody titers to fowl cholera vaccines and increased daily mortality (up to 3% in 12-week-old chicks) [89]. Aflatoxin consumption increases susceptibility not only to fowl cholera but also to Candida, Marek’s disease, paratyphoid infections, and more, due to its broad immunosuppressive effects [93].

Infectious bursal disease (IBD)

IBD and aflatoxicosis frequently co-occur, creating a feedback loop of immunosuppression. A study in Nineveh Province (Iraq) found a high correlation between IBD and aflatoxin B_1_ contamination in broiler farms. Although the aflatoxin levels were low (0.186–0.23 ppb), affected birds had reduced IBD antibody titers and severe lesions (hydropericardium, liver pallor, kidney necrosis), suggesting that even trace levels of AFB_1_ can exacerbate viral pathogenesis [94].

Infectious laryngotracheitis (ILT)

In an experimental model, broilers exposed to 200 ppb of AFB_1_ and vaccinated for ILT showed significantly lower antibody titers, reduced total serum proteins, and elevated liver enzymes. Histopathology confirmed ILT lesions in tracheal tissues. Withdrawal from AFB_1_ partially restored immunity, confirming the suppressive effect of aflatoxins on vaccine-induced protection [73].

Newcastle disease virus (NDV) and infectious bronchitis virus (IBV)

Broilers fed diets contaminated with DON and AFB_1_ show lower NDV and IBV antibody titers, which compromises herd immunity and increases disease risk. The humoral immune response fluctuates based on the mycotoxin dose and duration of exposure, and is further influenced by vaccination schedule and maternal antibody levels [43,74].

## 5. Implications for Consumer Health

The effects of mycotoxins extend beyond poultry production and into the food supply chain, directly affecting human health through the consumption of contaminated meat and eggs. As broiler chickens serve as a primary protein source in many regions, understanding and mitigating the risks of mycotoxin residues in poultry products is essential for public health protection.

### 5.1. From Feed to Food: The Contamination Pathway

Mycotoxins introduced via feed can persist through the digestive tract and undergo biotransformation in the liver of poultry. These metabolites—including aflatoxin M_1_, aflatoxicol, ochratoxin derivatives, and fumonisin analogs—can accumulate in edible tissues such as liver, muscle, and eggs. This contamination is not always eliminated by cooking, freezing, or industrial processing, which increases the risk of chronic exposure in human consumers [24,26].

Feed contamination acts as the initial point of entry, and poor storage practices, high humidity, and improper moisture control exacerbate the fungal growth responsible for mycotoxin production. Once integrated into the animal’s metabolism, these toxins are distributed systemically and may be detected in tissues for several days post-exposure [23].

### 5.2. Residue Accumulation and Food Safety Risks

Multiple studies have confirmed the presence of mycotoxin residues in broiler products. Aflatoxins and their metabolites are frequently found in liver and muscle samples, particularly in birds fed diets exceeding regulatory thresholds. OTA is known for its long half-life and strong protein-binding ability, contributing to its accumulation in both muscle and reproductive tissues [32,66].

One study found detectable AFM_1_ in both liver and muscle tissues when broilers were fed ≥200 ng/g AFB_1_. At lower concentrations (100 ng/g), AFM_1_ was still detected in liver tissues but not muscle, demonstrating a tissue-specific distribution pattern [24].

The consumption of such contaminated products is particularly dangerous for vulnerable populations, including infants and young children, who have lower detoxification capacity and higher food intake per body weight, pregnant and lactating women, where mycotoxins may cross the placenta or enter breast milk, immunocompromised individuals, who may be more susceptible to hepatotoxic, nephrotoxic, or carcinogenic effects, and chronic low-level exposure, that has been associated with increased risks of liver and kidney cancers, reproductive dysfunction, immunosuppression, and growth retardation [16,48].

### 5.3. Economic and Public Health Implications

Mycotoxin contamination has both direct and indirect economic consequences on (1) public health costs, by increased disease burden due to long-term toxin exposure, (2) trade barriers, by export restrictions and rejections based on residue testing failures, and (3) market instability, by consumer distrust and product recalls during outbreaks.

Producers may suffer heavy losses due to culling, feed recalls, or the need to implement costly detoxification strategies. National economies, particularly those reliant on poultry exports, may experience disruptions due to non-compliance with international food safety standards.

### 5.4. A One Health Perspective

The intersection of feed safety, animal health, and consumer well-being emphasizes the need for a One Health approach to mycotoxin control. Preventing mycotoxin contamination at the source—during crop harvesting, storage, and feed formulation—serves not only to enhance broiler productivity but also to reduce the burden of foodborne exposure in humans [95].

Investment in surveillance systems, harmonized global regulations, and preventive education for producers is a crucial component of this approach. A coordinated strategy linking animal and public health sectors can ultimately strengthen food system resilience.

Furthermore, mycotoxin contamination must be viewed not only as a food safety issue but also as an environmental and socioeconomic challenge. Climate change, for instance, is altering fungal growth patterns and toxin production globally, increasing the risk of contamination in regions previously considered low-risk. This dynamic underscores the need for adaptive, region-specific policies that integrate climate forecasting with mycotoxin surveillance in crops and animal feed [96].

In addition, the human health implications of mycotoxins extend beyond direct dietary exposure. Farm workers and feed handlers may be exposed to airborne mycotoxins in poorly ventilated environments, which can result in respiratory and dermal complications. Addressing these occupational risks requires collaboration between animal health experts, medical professionals, and environmental agencies. Promoting a One Health approach ensures that these diverse but interconnected risks are managed holistically—ultimately contributing to more sustainable livestock production and healthier communities [97].

## 6. Mitigation Strategies and Regulatory Considerations

Controlling mycotoxin contamination in broiler production is a multifaceted challenge that requires both preventive and corrective actions. These strategies must operate throughout the entire feed and production chain, from raw material handling and feed formulation to animal-level interventions. The goal is not only to protect broiler health and performance but also to ensure the safety of animal-derived food products for human consumption.

### 6.1. Regulatory Standards and Implementation Challenges

More than 100 countries have established maximum allowable levels for major mycotoxins in food and feed. The European Union has among the most stringent regulations, limiting AFB_1_ in food to 2.0 µg/kg and OTA to 5.0 µg/kg, with corresponding limits in animal feed [26,56].

However, enforcement and compliance are inconsistent across global regions. In many low- and middle-income countries, routine surveillance is limited or absent, and regulatory limits are either outdated or poorly enforced. Recent reports from Africa and Southeast Asia indicate frequent violations of safe limits in maize-based poultry feed and finished poultry products [8,98].

Furthermore, the co-occurrence of multiple mycotoxins is rarely accounted for in regulatory risk assessments, despite evidence that such combinations may exert synergistic or additive effects in both animals and humans [99]. These global disparities in food safety oversight are closely tied to the regulatory standards applied to animal feed, as outlined below.

The establishment of regulatory limits for mycotoxins in animal feed is a cornerstone of risk management. Countries such as those in the European Union have implemented strict thresholds for major mycotoxins, including a maximum of 20 µg/kg for aflatoxin B_1_ in poultry feed and guidance levels for DON and fumonisins. These regulatory frameworks are based on toxicological data and are intended to minimize the carry-over of residues into meat and eggs, and to preserve animal health [25,56].

However, a key challenge lies in the disparity of enforcement and monitoring capacity between countries. In many regions, especially in Africa and Southeast Asia, contaminated feed remains a persistent issue due to inadequate storage conditions, limited testing infrastructure, and variable climate patterns that favor fungal growth [8,98]. Even in countries with established regulations, enforcement may be sporadic or poorly resourced. As a result, the practical implementation of feed safety standards is often inconsistent, leading to chronic exposure in flocks and downstream effects on food safety.

### 6.2. Monitoring Feed and Ingredients

Effective mitigation starts with the detection of mycotoxins in feed ingredients before they reach the birds. Analytical techniques such as enzyme-linked immunosorbent assay (ELISA) and high-performance liquid chromatography (HPLC) are routinely used to screen for toxins. ELISA offers a cost-effective and rapid method that is suitable for field-level monitoring, with a sensitivity high enough to detect common mycotoxins like aflatoxins, DON, and fumonisins. HPLC, while more accurate and specific, requires advanced laboratory infrastructure and is less accessible in low-resource settings [18,100,101].

Regular monitoring programs should be part of integrated quality assurance systems within feed mills and farms. Feed manufacturers that source raw materials globally face higher risks of variable contamination. Without systematic screening and control, even small amounts of contaminated ingredients can compromise entire feed batches, particularly when combined with other stressors like high ambient temperatures or pathogen load.

### 6.3. Feed Additives and Experimental Detoxification Strategies

One of the most common mitigation approaches at the farm level is the use of mycotoxin binders—substances that are added to feed to reduce the bioavailability of toxins in the gastrointestinal tract. These agents work by adsorbing mycotoxins and preventing their absorption into the bloodstream.

Among the most widely used are clay-based binders such as bentonite, zeolite, and hydrated sodium calcium aluminosilicate (HSCAS). These materials are particularly effective against aflatoxins due to their polarity and molecular structure, which allows stable binding in the acidic conditions of the stomach. However, their efficacy against less polar toxins like DON or fumonisins is limited [102]. Adsorbents such as bentonite and yeast cell wall extracts have shown partial efficacy in restoring antibody levels when included in the diet alongside mycotoxins [75].

To broaden the spectrum of protection, feed producers increasingly use compound binders that combine mineral adsorbents with biological materials like yeast cell wall extracts. These biological components can adsorb a wider variety of toxins and may also support gut health and immune responses. While the evidence for such compounds is growing, results can vary depending on the formulation and the specific toxin profile present in the feed.

Numerous experimental studies have evaluated the efficacy of various mycotoxin binders and detoxifiers under controlled and field conditions. These interventions aim to reduce mycotoxin absorption, support detoxification pathways, and mitigate physiological damage caused by chronic exposure.

The fundamental mechanism of most binders is the formation of non-resorbable complexes with mycotoxins in the gastrointestinal tract, thereby reducing their bioavailability and systemic distribution. Clay-based adsorbents such as bentonite, zeolites, and hydrated sodium calcium aluminosilicate (HSCAS) are widely used for their ability to bind aflatoxins and some trichothecenes. A comparative study concluded that aluminosilicates like zeolites and HSCAS were more efficient than some commercial detoxifying agents, particularly against aflatoxins [102].

The administration of binders alongside antibiotics or coccidiostats has raised concerns about potential interference with drug absorption. However, a pharmacokinetic study showed that the co-administration of binders with antimicrobials like diclazuril and salinomycin did not significantly alter their bioavailability or absorption profiles in broiler chickens [103,104].

Performance-related outcomes in broilers fed mycotoxin-contaminated diets with or without adsorbents have shown variable results. One study reported that broilers receiving bentonite-based binders exhibited reduced weight gain during early growth but no significant differences in body weight or organ weights at the end of the trial. Notably, no mycotoxin residues were detected in the liver or breast muscle, suggesting effective binding and elimination [105]. Another trial revealed liver function benefits when binders were used, evidenced by reduced serum liver enzymes and fewer histopathological alterations compared to control birds [106].

The integration of natural components, such as yeast extracts, prebiotics, and plant-derived polyphenols, has shown synergistic effects. In broiler breeders exposed to ochratoxin A, combinations of clay minerals and these biologically active additives led to improvements in laying performance, egg quality, organ weight, and bone mineralization [66].

Enzymatic detoxifiers, which biochemically transform mycotoxins into less toxic metabolites, have also been evaluated. In one experiment, broilers fed with enzymatic detoxifier-supplemented feed exhibited reduced severity of hepatic and renal lesions typically associated with aflatoxicosis. Observed pathology included reduced subcapsular hemorrhages, lower fat accumulation in the liver, and improved kidney architecture [84].

Several studies have also explored the efficacy of novel commercial blends. Products like HASCS, Propower, and AVI5 bac have demonstrated protective effects in broilers exposed to ochratoxins. Treated groups showed lower mortality rates, improved weight gain, and reduced clinical signs such as diarrhea and ruffled feathers. These benefits were accompanied by improvements in serum biochemistry, including reduced levels of ALT, bilirubin, uric acid, and restoration of immunoglobulin levels [107].

The use of antioxidant-rich botanical compounds has gained traction as a supportive strategy. Silymarin, a flavonoid complex derived from Silybum marianum (milk thistle), has demonstrated hepatoprotective and immunomodulatory properties in multiple poultry studies. It acts by stabilizing hepatocyte membranes, scavenging free radicals, and promoting the synthesis of protective proteins. Supplementation with silymarin has been associated with reduced hepatic enzyme levels, improved antioxidant capacity, and overall enhancement of liver function under mycotoxin stress [71].

Together, these findings support the strategic use of mycotoxin binders and detoxifiers as practical tools in poultry production. While efficacy can vary depending on the type and level of mycotoxin exposure, formulation, and bird age, such interventions remain critical components of integrated mycotoxin management programs.

### 6.4. Antioxidants and Nutritional Modulators

Mycotoxins exert part of their toxic effect through oxidative stress and inflammation [103]. As a result, dietary supplementation with antioxidants can help mitigate some of the damage. Vitamin E, for example, has been shown to reduce the immunosuppressive and hepatotoxic effects of ochratoxin A by stabilizing cell membranes and improving antioxidant enzyme activity [70].

In addition to vitamins, certain herbal compounds like silymarin, a flavonoid extracted from milk thistle, have demonstrated hepatoprotective effects. Silymarin stabilizes hepatocyte membranes, scavenges free radicals, and modulates cytokine responses. Its use in mycotoxin-challenged birds has resulted in improved liver function, reduced serum enzyme levels, and better feed conversion ratios [71].

### 6.5. Biological Interventions and Probiotics

Biological strategies are also gaining attention. Some probiotic strains, particularly *Lactobacillus salivarius* and *Lactobacillus plantarum*, have demonstrated the ability to degrade aflatoxins or reduce their absorption in the intestine. These probiotics may also enhance gut barrier function and modulate immune responses, offering dual benefits under mycotoxin stress [80].

While these results are promising, the variability in probiotic efficacy between strains and conditions makes standardization challenging. Further research is needed to validate these approaches under commercial conditions and to determine optimal dosages and combinations.

### 6.6. Prevention Through Crop and Storage Management

Despite the importance of feed additives and dietary interventions, prevention at the crop level remains the most effective long-term strategy. The implementation of Good Agricultural Practices (GAPs)—including timely harvesting, proper drying, pest control, and storage under dry, cool conditions—can drastically reduce fungal contamination.

Similarly, Good Storage Practices (GSPs) help prevent post-harvest fungal growth. Regular aeration, moisture control, and use of antifungal agents can minimize the conditions that allow mycotoxins to proliferate during storage and transport. These practices are particularly critical in tropical regions where temperature and humidity promote fungal activity year-round.

## 7. Conclusions and Prospects

Mycotoxins remain a pervasive and complex threat to poultry production, affecting not only broiler growth and health but also compromising vaccine efficacy and food safety. Their diversity, environmental factors contributing to their presence, and their potential to act synergistically with pathogens highlight the urgent need for integrated control strategies across the poultry value chain.

This review followed a two-part structure: an overview of the main mycotoxins, followed by their broader impact on broiler physiology, immunity, and public health. This format was designed to provide both a detailed account of individual toxins and a thematic synthesis.

Chronic exposure to aflatoxins, trichothecenes, DON, fumonisins, ochratoxin A, and zearalenone, even at subclinical levels, can weaken immune responses, impair gut function, and reduce vaccine efficacy, increasing susceptibility to infections such as coccidiosis, salmonellosis, *E. coli*, and viral diseases. The transfer of residues into meat and eggs also raises serious concerns for food safety, especially in regions with weak regulatory enforcement.

Effective mitigation requires a combination of feed monitoring, good agricultural and storage practices, and the use of binders and detoxifying agents. Nutritional and biological interventions, including antioxidants, probiotics, and plant-derived compounds, can reduce the effects of toxins and support animal resilience.

Managing mycotoxins in broiler systems is not only critical for productivity but also central to protecting human and animal health under a One Health framework. Continued research, especially on vaccine–mycotoxin interactions and region-specific control strategies, will be essential to strengthen food system safety and sustainability.

Future research should prioritize longitudinal studies evaluating the cumulative effects of multi-mycotoxin exposure and their impact on vaccine efficacy under commercial production conditions. Innovations in real-time feed monitoring, genomic resistance to mycotoxin damage, and integrated detoxification technologies also hold promise for reducing the burden of mycotoxins across poultry systems. Emphasis on region-specific challenges, including climate-driven fungal proliferation and limited regulatory infrastructure, can further inform targeted interventions and policy development.

## Figures and Tables

**Figure 1 toxins-17-00261-f001:**
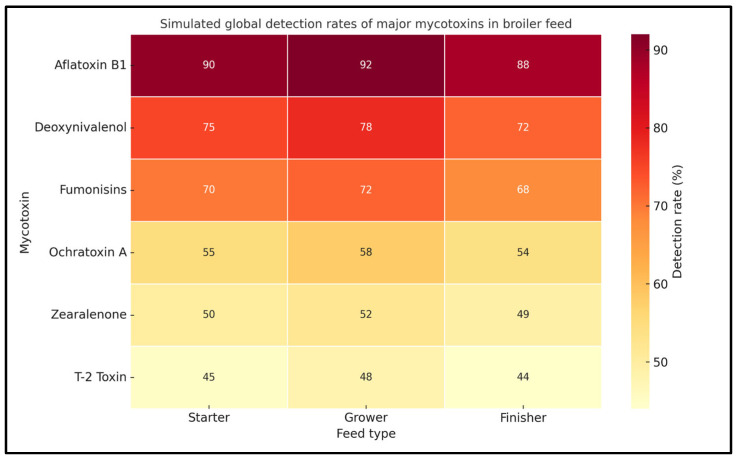
Simulated global detection rates of major mycotoxins in broiler feed.

**Figure 2 toxins-17-00261-f002:**
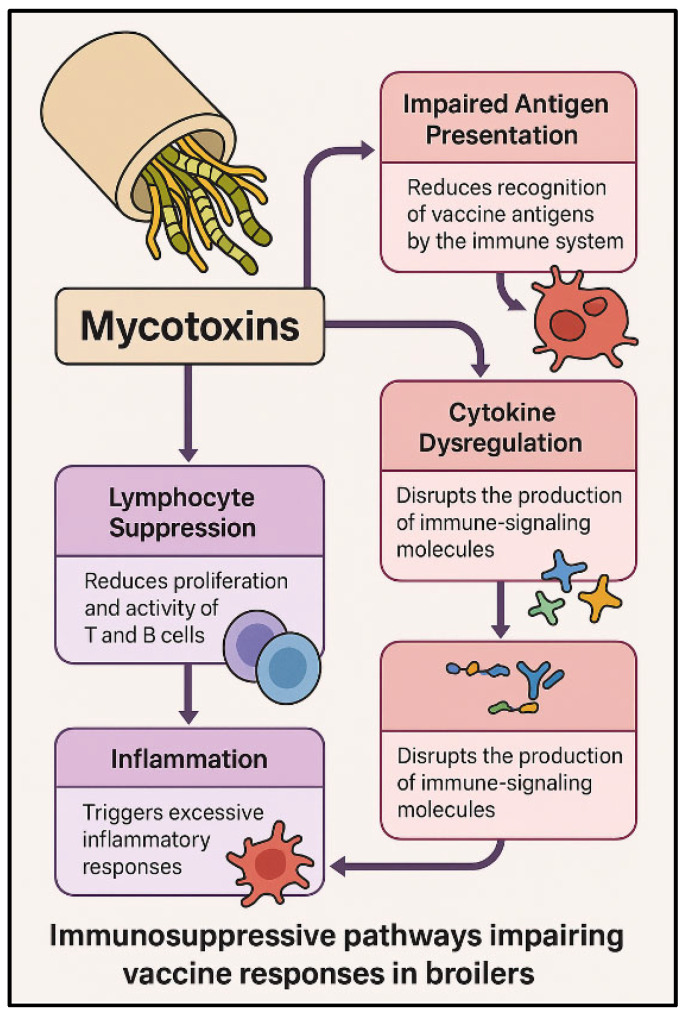
Immunosuppressive pathways impairing vaccine responses in broilers exposed to mycotoxins. Mycotoxins affect the immune system through multiple mechanisms, including impaired antigen presentation, cytokine dysregulation, and lymphocyte suppression. These alterations result in weakened immune signaling, reduced T and B cell activity, and excessive inflammatory responses, ultimately compromising vaccine efficacy.

**Table 1 toxins-17-00261-t001:** Overview of major mycotoxins in broiler feeds, including their main fungal producers, target organs or systems in chickens, and key pathological and physiological effects. This summary supports a clearer understanding of how different mycotoxins influence poultry health, productivity, and immune function—information that is essential for effective risk management and mitigation strategies in broiler production.

Mycotoxin	Main Fungal Producers	Target Organs/Systems	Key Effects
Aflatoxins	*Aspergillus flavus*, *A. parasiticus*	Liver, kidney, immune organs	Hepatotoxicity, immunosuppression, carcinogenicity
Trichothecenes	*Fusarium* spp., *Myrothecium*, *Stachybotrys*	Liver, immune system	Protein synthesis inhibition, immunosuppression
Deoxynivalenol (DON)	*Fusarium graminearum*	GIT, immune system, liver	GIT damage, immunomodulation, feed refusal
Fumonisins	*Fusarium verticillioides*, *F. proliferatum*	Liver, kidneys, immune system	Hepatic necrosis, immune disruption, poor growth
Ochratoxin A (OTA)	*Aspergillus* spp., *Penicillium* spp.	Kidneys, liver, immune system	Nephrotoxicity, immunosuppression, embryotoxicity

## Data Availability

No new data were created or analyzed in this study. Data sharing is not applicable to this article.

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
