# Peer review of "Mycotoxins in Broiler Production: Impacts on Growth, Immunity, Vaccine Efficacy, and Food Safety"

_toxins, 2025, doi:10.3390/toxins17060261_

Round 1
Reviewer 1 Report
Comments and Suggestions for Authors
Dear Authors,
This study reviewed the mycotoxins in broiler production including the impacts on chicken growth, immunity, vaccine Efficacy, and food safety. Mycotoxins remain a pervasive and complex threat to poultry production, affecting not only the growth performance and health of broiler chickens but also compromising the efficacy of vaccination programs and the safety of animal-derived food products. The diversity of mycotoxin types, their environmental and geographic variability, and their potential to act synergistically with pathogens underscore the urgent need for integrated management strategies across the poultry value chain. This review highlights the multifactorial effects of common mycotoxins on broiler physiology, immune competence, and disease resistance. The manuscript was well written. Before the manuscript was accepted to be published in the journal Toxins, some places need to be revised.
(1) Section 2 (Main types of mycotoxins relevant in broiler feeds) introduced the background and general knowledge of mycotoxins. I suggest shortening this section and merging it with the section 1 (Introduction).
(2) Line 109, “…by Aspergillus flavus and Aspergillus parasiticus [16]” should be revised as ““…by Aspergillus flavus and A. parasiticus [16]”
(3) The title of Section “7. Conclusion” should be revised as “7. Conclusion and Prospects”. The title of the subsection “7.1 Future Directions” should be deleted.
Author Response
Dear Reviewer,
We would like to sincerely thank you for your thorough evaluation of our manuscript and the constructive comments provided. We are grateful for the positive feedback on the quality of the writing and the relevance of our review on mycotoxins in broiler production. We have carefully considered each suggestion and made the recommended revisions to improve the manuscript's clarity, structure, and overall quality. Below, we provide a detailed, point-by-point response to each comment, outlining the changes made.

Reviewer 2 Report
Comments and Suggestions for Authors
The topic is of interest and according to the aims of the journal. However, the following comments/ suggestions should be incorporated.
Introduction
Line 63-64: I think 450 or more mycotoxins are known, please check
Contents
The contents of the review article should be more specific in one direction.
The present review has a mixture of many different types of mycotoxins. Please specify which kind of mycotoxins are mainly occurring in animal feed samples; similarly, more attractive figures should be provided.
Comments on the Quality of English Language
English is acceptable
Author Response
Dear reviewer,
We sincerely thank you for the thoughtful and constructive feedback. We appreciate the recognition of the manuscript's relevance to the aims of the journal and have carefully considered all comments. In response, we have revised the manuscript to address the concerns raised, including updating factual content, clarifying the focus of the review, and enhancing visual elements. Below is a detailed, point-by-point response outlining the changes made to improve the clarity, specificity, and overall quality of the manuscript.

Reviewer 3 Report
Comments and Suggestions for Authors
Dear Authors,
in your review paper, you describe mycotoxins in chickens, their impact on growth, immunity, public health and the concept of One Health. The positive side is the topic itself, which is very relevant and useful for many researchers dealing with poultry, and the negative is the concept of the review itself, which differs in first and second parts and makes it difficult to understand.
General comments: you should decide if the review will be organized according to the mycotoxins (as is from L 107-374) or according to the objectives of this review (growth, clinical signs, pathology, immunology, impact on human health, One Health perspective, etc.). For example, currently in chapter 2. Main types of mycotoxins relevant in broiler feeds (L 107) there are more data on clinical/pathological findings and immunology then on mycotoxins in broiler feed (in this form, the text on specific mycotoxin contain all the data related to the objectives of the review)
Specific comments:
L 56-57 please rephrase, not clear enough
L 75-79 maybe to relocate, after L 58?
L 54-85 and L90-92 what is the difference?
L 106 please change the title (main types?)
What were you guided by when you decided on the arrangement of listing mycotoxins in this chapter? pathogenicity, the alphabet, something else?
L 114-118 and then later on you have a whole chapter on this topic?
L 215-218 this part is more suitable for chapter 6.3
L 311-317 more suitable for chapter 6.4
L 355-360 this is a description of the table 1, so it should be written under the table, as a subtext?
L 363-374 this is a description of the figure 1, so it should be written under the figure, as a subtext? Please mention the sources of the data included in the Figure 1
L 405 writing of Gallinarum
L 405-407 this sentence is more suitable for other chapter
Chapters 3.3. and 3.4. should be extended, if there are no data related to poultry/chickens, readers will for sure valid positive the data related to other animals or humans
L 456 first write in full and then abbreviation
L 468 and 477 please uniform (first pathogen then diseases or vice versa)
Chapter 4.3 please combine viral and then bacterial diseases
L 553-563 and L 578-581 please write without the bullets
L 582-585 please mention references for those statements
Chapter 5.5 should be extended, currently is in the form of a conclusion?
Chapters 6.1 and 5.3 – could be combined?
Chapters 6.3 and 6.7 could be combined (actually it seems as chapter 6.7 combine all previous, from 6.1 to 6.6.?
Author Response
Dear reviewer,
We would like to express our sincere thanks for the thoughtful and comprehensive evaluation of our manuscript. We are especially grateful for the recognition of the topic’s relevance to the poultry and One Health research communities.
We have carefully reviewed all general and specific comments, and we greatly appreciate the clarity and depth of the feedback provided. These observations have helped us identify areas where the manuscript’s structure, coherence, and focus can be improved. In response, we have undertaken several revisions to enhance the organization and readability of the review, particularly by clarifying the thematic structure, refining section headings, and relocating certain content for better alignment with the objectives of the manuscript.
We have addressed each of the reviewer’s comments point by point below, and we trust that the revised version will now offer a more cohesive and accessible overview of the topic for a broad readership.
We have added a new figure and eight reference titles, and made revisions throughout the manuscript according to the reviewers' suggestions using track changes.

Round 2
Reviewer 1 Report
Comments and Suggestions for Authors
Dear Authors,
The manuscript was revised. I have no further comments, and recommend the manuscript to be accepted in its current form.
Reviewer 3 Report
Comments and Suggestions for Authors
Dear Authors, thank you for providing your comments and taking into account all the suggestions.